# Conservation and Global Distribution of Onion (*Allium cepa* L.) Germplasm for Agricultural Sustainability

**DOI:** 10.3390/plants12183294

**Published:** 2023-09-18

**Authors:** Kingsley Ochar, Seong-Hoon Kim

**Affiliations:** 1Council for Scientific and Industrial Research, Plant Genetic Resources Research Institute, Bunso P.O. Box 7, Ghana; kingochar@yahoo.com; 2National Agrobiodiversity Center, National Institute of Agricultural Sciences, Rural Development Administration, Jeonju 5487, Republic of Korea

**Keywords:** onion, diversity, germplasm, genebank, climate change

## Abstract

Onion (*Allium cepa* L.) is recognized globally as a crucial vegetable crop, prized not only for its culinary applications but also for its numerous health-promoting properties. With climate change relentlessly exerting mounting challenges to agriculture, the preservation and deployment of onion germplasm has become critical to ensuring sustainable agriculture and safeguarding food security. Global onion germplasm collections function as repositories of genetic diversity, holding within them an extensive array of valuable traits or genes. These can be harnessed to develop varieties resilient to climate adversities. Therefore, detailed information concerning onion germplasm collections from various geographical regions can bolster their utility. Furthermore, an amplified understanding of the importance of fostering international and inter-institutional collaborations becomes essential. Sharing and making use of onion genetic resources can provide viable solutions to the looming agricultural challenges of the future. In this review, we have discussed the preservation and worldwide distribution of onion germplasm, along with its implications for agricultural sustainability. We have also underscored the importance of international and interinstitutional collaboration in onion germplasm collecting and conservation for agricultural sustainability.

## 1. Introduction

The genus *Allium* is one of the most widespread monocotyledonous plants (Figure 1), constituting over 800 plant species [1,2]. The genus includes a multitude of widely cultivated forms such as onion (*Allium cepa* L.), chive (*Allium schoenoprasum* L.), green onions or Japanese bunching onion (*Allium fistulosum* L.), garlic (*Allium sativum* L.), shallot (*Allium ascalonicum* L.), scallion (*Allium wakegi*), and leek (*Allium ampeloprasum* L. var. *porrum*) [1,2,3]. Within thisgenus, onion (*Allium cepa*; Family:Alliaceae),a diploid species with 16 chromosomes (2n = 2x = 16), is the most extensively cultivated and consumed vegetable crop [4]. It ranks second only to tomatoes in terms of global production [5,6,7]. With a long and varied history, onion is believed to be among the first plant species to be domesticated during the era of ancient human civilization [8,9]. For instance, archaeological evidence suggests that onion domestication took place over 4000 years ago [10,11,12]. The domestication process, like in many other crop species, involved cultivating wild onion species and selecting specific genotypes based on desirable characteristics determined by early growers [3].

The practice of onion cultivation and its subsequent worldwide spread isthought to have closely followed human migration patterns and trade routes [3,13]. Owing to their unique flavor, health-promoting properties, and culinary versatility, onions have become a staple in human diets and cultures over millennia, adding flavor and aroma to countless dishes [14,15]. Presently, onion cultivation is distributed over more than 140 countries. The largest global producers of onions include China, India, the United States, and Turkey [16]. Additionally, high onion productivity has been recorded in countries such as the Republic of Korea (73.21 t/ha) and Guyana (77.75 t/ha) [10]. The diversity of onion cultivars varies across different geographical locations and agroclimatic conditions, influenced by factors such as climate and soil suitability, as well as consumer or market preferences [17,18]. The edible components of the onion plant, encompassing both green foliage and the bulb, are treasured not only for their culinary applications but also for their nutritional and medicinal properties [8,19].

In the current climate, characterized by evolving global climate variables and myriad environmental challenges, both plant species and our agricultural systems are increasingly vulnerable, with potentially severe ramifications for global food security [20,21]. As a crop of significant global importance, the process of gathering, preserving, and utilizing the germplasm of the onion becomes an indispensable strategy to prevent the loss of crucial genes and gene pools [22]. Germplasm is a long-term resource management operation and provides knowledge about species genetics, and this is useful for conserving plant diversity and achieving both food and nutritional security [22]. By conserving onion germplasm, the genetic diversity of the crop is preserved and further used in developing new resilient onions with improved traits, such as drought tolerance, disease resistance, and adaptability to changing climatic conditions and promoting long-term agricultural sustainability [23,24]. Research has revealed that nearly 7.4 million plant accessions have been gathered and conserved in approximately 1750 plant germplasm centers globally [25]. Yet, only a minuscule 2% of these resources have been utilized as Plant Genetic Resources (PGRs) [26,27]. Worldwide, an abundance of germplasm collections serves as repositories for genetic resources. In onion, these collections encompass a broad spectrum of onion varieties, landraces, and wild relatives, each representing unique genetic traits and adaptations. Conservation initiatives that focus on the maintenance, characterization, and documentation of onion germplasm are pivotal in preventing genetic erosion and loss. As is the case with other crop species, onion germplasm collecting could be effective under research partnership agreements among international organizations, genebanks, and research institutions [28]. These partnerships ensure the expansion and long-term conservation of the crop’s rich diversity. This collaborative framework enables researchers and breeders to access a wide array of onion germplasm from diverse geographical regions, which is crucial for the development of new varieties [28]. International networks and platforms facilitate the exchange of germplasm, foster collaboration, and propel research in genetic improvement. Through meticulous breeding and selection processes, breeders can incorporate desirable traits from diverse onion germplasm into commercially viable onion varieties [29]. This procedure entails identifying onions with particular traits, including drought tolerance, heat tolerance, water-use efficiency, disease and pest resistance, and adaptability to specific climates [30,31]. By cultivating climate-resilient onion varieties, farmers can offset the detrimental impacts of climate change on onion production, bolster yield stability, and foster agricultural sustainability. Climate-adapted onion varieties reduce farmers’ vulnerability to fluctuating climatic conditions, ensuring a steady onion supply and, thereby, enhancing food security. Resilient onion varieties, developed from diverse germplasm resources, aid in decreasing dependence on agrochemical inputs like pesticides and fertilizers by integrating natural resistance to pests and diseases [32]. In general, collaboration and knowledge sharing among germplasm collection institutions and researchers worldwide spark innovation and facilitate the development of region-specific onion varieties. Furthermore, effective conservation and global distribution of onion genetic resources areessential in advancing agricultural sustainability and onion production in the face of climate change. The current review aims to provide a current overview of the conservation and global distribution of onion germplasm as a crucial component for sustaining agriculture and ensuring food security.

## 2. Onion Genetic Resources

The practice of cultivating onions dates back to ancient times, when growers in various regions selectively propagated onion populations exhibiting desirable traits [7]. This process has resulted in a broad array of landraces and cultivars, each endowed with significant genetic variation [10,29]. Meanwhile, the unselected populations have persisted in the wild, serving as wild relatives of the cultivated types. Onion genetic resources comprise diverse collections of onion varieties, landraces, and wild relatives, each carrying a plethora of genetic traits [33,34]. Generally, onion germplasm collection shows variation in several traits, including yield perplant, shape index, bulb dry weight, bulb hardiness, bulb height, bulb diameter, the color of dry skin, dry skin thickness, dry matter ratio, number of leaves, and leaflength [35,36,37,38]. Thus, the diversity of onion resources ismeticulously preserved to ensure the maintenance and utilization of their abundant genetic diversity for a range of purposes, including breeding programs, advancing research, and promoting conservation initiatives [39].

### 2.1. Onion Wild Relatives and Landraces

Onion crop wild relatives and landraces are invaluable constituents of onion germplasm collections, especially their pungency, flavouring, and traditional medicinal value;they represent an excellent resource for breeding new cultivars [28,40]. They epitomize the genetic diversity and ancestral forms of cultivated onions. Both wild relatives and landraces contribute significantly to the conservation and utilization of onion genetic resources for breeding, thus promoting agricultural sustainability. They provide a vast reservoir of genetic diversity, encompassing unique adaptive traits [41,42]. By conserving and leveraging these genetic resources, we can ensure the sustained advancement and resilience of onion cultivation. In doing so, we enhance food security and promote the sustainable utilization of plant genetic resources [43,44].

#### 2.1.1. Onion Wild Relatives

Crop wild relatives are species closely related to cultivated plants, including onions. Having evolved in a variety of ecological niches, these wild onion relatives often exhibit adaptations to specific environmental conditions [3]. They display variations in traits such as disease resistance and tolerance to environmental stressors, traits that may be absent in cultivated varieties [3]. These traits can be incorporated into breeding programs to augment the adaptability and resilience of cultivated onions [45,46,47]. For instance, certain wild onion species such as *A. asarense*, *A. roylei*, *A. galanthum*, *A. oschaninii*, *A. turkestanicum*, *A. pskemense*, *A. altaicum*, *A. farctum*, *A. praemixtum*, *A. rhabdotum*, *A. pskemense*, and *A. vavilovii* demonstrate tolerance to harsh conditions such as drought, extreme temperatures, or poor soil [18]. Phylogenetic studies have revealed *A. vavilovii* as the most closely related wild relative of the common onion [48,49]. By studying the crossabilityand integrating adaptive traits of these wild relatives, breeders can create onion varieties better equipped to endure stressful growing conditions [3,50]. The presence of genetic diversity in wild relatives aids in broadening the gene pool available for onion breeding programs [51]. The introduction of genes from these wild relatives can enhance traits like disease resistance, nutritional quality, and flavor [40,52]. These genetic resources can contribute to the development of improved onion varieties that meet specific consumer and market demands, as well as tackle emerging challenges in onion cultivation.

#### 2.1.2. Landraces of Onions

Landraces are traditional, genetically heterogeneous varieties that have evolved and adapted to specific ecogeographical locations, typically conserved by farmers over generations [42,53]. Integral to local cultures or traditions, landraces often offer historical and cultural benefits, exemplifying the culinary heritage of certain regions [54]. These genetic resources have evolved to thrive in particular local conditions, including specific climates, soil types, and farming practices [55]. Over centuries of cultivation, onions have adapted to a range of conditions, such as temperature and photoperiods, leading to the emergence of a diverse array of landraces [56]. These onion landraces have developed the ability to withstand a variety of stressors, including pests, diseases, and several abiotic environmental factors prevalent in their native regions [32]. The conservation of onion landraces, including other types of genetic resources, not only helps maintain the knowledge associated with traditional onion cultivation but also contributes to the preservation of cultural diversity [57]. Many landraces may possess unique flavors and storability, which renders them valuable sources of genetic diversity [58]. By conserving and utilizing onion landraces, farmers can maintain a diverse range of onion varieties that are well-suited to local conditions. This not only contributes to local food security but also reduces dependence on a limited number of commercial cultivars, thus enhancing profitability. Through careful characterization, evaluation, and crossbreeding with cultivated onions, the genetic diversity of wild relatives and landraces can be harnessed to develop improved onion varieties [4]. This approach can help address emerging challenges in agriculture, improve the nutritional content of crops, and meet specific consumer preferences [3].

### 2.2. Onion Cultivars and Breeding Lines

Onion cultivars and breeding lines are the tangible outcomes of continuous efforts by breeders to produce improved varieties that exhibit desirable traits and characteristics apt for commercial cultivation. This progress is largely facilitated through selective breeding and hybridization techniques [8]. In the context of onion breeding, specific objectives may include enhancing traits such as yield potential, flavor profile, storage longevity, adaptability to diverse environmental conditions (resistance to various diseases and pests), and bulbcharacteristics (color, shape, soluble-solids content, pungency and flavor, storage ability, andhealth-enhancingattributes) [3]. The development and proliferation of such improved cultivars contribute significantly to optimizing agricultural productivity, profitability, and sustainability.

#### 2.2.1. Commercial Varieties

Commercial cultivars represent the culmination of extensive breeding and selection processes optimized for desirable traits to ensure high yields and market acceptance. Widely cultivated by farmers for their market viability, commercial onion cultivars are selected based on consumption preferences and market demands [59]. Key selection criteria often include size, shape, pungency, color, storage qualities, resistance to pathogens, pests, and bolting [3,29]. Commercial cultivars bred specifically for high-yielding and quality seed production for the market must have desirable attributes such as vigorous bulbs, with flowering uniformity straight seed stalks, number of umbels produced, and the number of flowers per umbel [3]. Moreover, there exist regionally adapted or specialty onion cultivars developed specifically to flourish under unique climate conditions, soil types, or cultural settings. These regional cultivars are particularly suited to local growing conditions and may exhibit distinct flavor profiles or other unique characteristics aligned with regional culinary preferences [56]. Such advancements significantly contribute to the progress of onion cultivation by addressing challenges, catering to the evolving needs of growers and consumers, and aligning with broader trends within the agricultural industry [59]. A prerequisite to success in breeding commercial onion cultivars is the presence and exploitation of diversity in germplasm collection.

#### 2.2.2. Breeding Lines

Breeding lines, or parental lines, serve as the foundation for developing new onion cultivars. Breeding programs aim to create genetically diverse and stable lines to serve as the basis for hybridization [3]. These lines are meticulously selected for specific traits of interest, such as disease resistance or unique flavor profiles [60]. The process of crossing different parental lines allows breeders to combine desirable traits from each parent, resulting in hybrids with distinctive and superior attributes, such as improved vigor, yield uniformity, or disease resistance, compared to their parent lines [3,45].

Within onion breeding programs, disease or pathogen resistance, alongside edible yield, is a key targeted trait [51]. Enhancing disease resistance reduces the reliance on chemical treatments, thus promoting sustainable onion production. In parallel, high-yield breeding lines are cultivated to boost onion productivity and optimize agricultural efficiency [32]. These lines are chosen based on their capability to produce a larger number of bulbs per unit area or exhibit other characteristics that contribute to increased yield potential [59]. Consequently, high-yielding lines can enhance farmers’ profitability by maximizing onion production.

## 3. Intraspecific Classification of Onion

The worldwide genepool of onion is estimated to be over 20,000, and this can be subdivided into three distinct groups: common group, aggregatum group, and ever-ready group [61]. Members of the common onion group comprise open-pollinated traditional and modern cultivars, hybrids, and local landraces under cultivation across the globe [62]. Thus, there is a wide diversity within bulbous or common onion groups and much variation across different countries [10,63]. This category of onions reproduces from seeds or from seed-grown sets and is normally made up of a single large bulb [64]. Many dry bulb onions under cultivation are members of this category. Common onions have high export value in many countries, such as India and China, and relative to the common onion group, the aggregatum group of onions is of less economic significance [65,66]. The aggregatum group of onions, also known as multiplier onions, reproduce mainly by asexual means, involving many small-sized daughter bulbs that form aggregated clusters [66,67]. Members of the common onion group are predominantly cultivated in regions including Asia, Europe, and America, specifically for their dry bulbs [65]. The aggregatum group of onions, also known as small onions or multiplier onions, possess excellent storage capacity [64,66]. Compared to common onions, the aggregatum group of onions ishardier and early maturing [65]. Shallots represent the most significant subgroup of the aggregatum group that is commercially cultivated [65,68]. Shallots produce numerous aggregated undersized bulbs that are narrowly ovoid to pear-shaped and usually havered-brown or coppery skins. The last category, the ever-ready onion group, consists of onions that are known for their prolific vegetative growth features, making them distinct from the other groups. They produce their leaves and bulbs throughout the year [69]. Around the mid-20th century, ever-ready onions were known to be commonly cultivated in Great Britain.

## 4. Ex Situ Conservation Efforts for Onion Genetic Resources

Maintaining onion genetic diversity is fundamentally essential for the sustainable development of improved onion varieties and for purposes of scientific research and conservation [32,70]. The safeguarding and utilization of these genetic resources through seed banks, conservation programs, and collaborative efforts underpin the continuous advancement and adaptability of onion cultivation amidst shifting agricultural and environmental conditions. Conservation efforts for genetic resources commonly employ both in situ and ex situ methods. The in situ conservation emphasizes the preservation of plant species within their natural habitats, thereby maintaining their ecological context and interactions with other organisms [71,72]. Conversely, ex situ conservation involves the cultivation and preservation of genetic resources in controlled environments (or outside the plant’s natural environment) such as tissue culture laboratories, botanical gardens, greenhouses, cryopreservation, or field genebanks [71]. In the domain of plant genetic conservation, these strategies complement each other, ensuring the preservation and availability of a wide range of plant genetic resources for future generations [72]. In the context of onion genetic resource conservation, ex situ conservation tends to be more suitable and represents the most commonly employed technique compared to the in situ approach.

### 4.1. Seed Genebanks

It is now clear the possibility of losing many plant species in the terrestrial ecosystem, especially amid the unabated changing climatic conditions and habitat loss [73]. In this regard, long-term storage of seeds of economically important crops in genebanks has become an indispensable component in the effort to promote agricultural sustainability [74]. Onions, as one of the globally recognized vegetable species, require conservation attention. Typically, onion collections encompass seeds sourced from various geographical regions, representing an extensive array of genetic diversity [75]. Preserving the genetic diversity of onion is essential for maintaining the crop’s resilience in the face of environmental challenges and evolving pathogens [76]. The conservation of onion seeds (orthodox seeds) in genebanks ensures the availability of diverse genetic materials for regional and global germplasm exchange and for future applications in research [57]. Since onion seeds, like other Allium seeds, haveshort longevity, to ensure their prolonged viability, the seeds are meticulously stored under tightly regulated conditions, including low temperature and humidity.

### 4.2. Tissue Culture or In Vitro Conservation of Onions

In vitro collections or plant tissue culture facilitate the conservation of onion genetic resources such as tissue cultures, embryos, or shoot meristem cultures [77]. This methodology entails the propagation of minute onion tissue pieces under sterile conditions using nutrient-rich media [78,79]. Advantages of in vitro onion collections include compact storage, diminished space requirements, and the capacity for rapid multiplication and distribution of plant materials, in particular vegetative propagated materials [80]. This method proves particularly beneficial for conserving onion varieties that are rare or endangered and *Allium* species that pose challenges to preservation through conventional seed storage techniques [81].

### 4.3. Cryopreservation

For long-term storage of onion germplasm resources in genebanks, the seed storage system is the most convenient and easiest. In recent years, the efficiency of onion germplasm management has seen a greater improvement, especially following the development of new methods such as cryopreservation of Alliums’ genetic resources [82]. Cryopreservation represents an advanced ex situ preservation method for long-term storage of germplasm, involving freezing at ultra-low temperatures (−196 °C) in liquid nitrogen media [9,78,83]. This technique has contributed to increased effectiveness in the management of many crop accessions stored in various genebanks, providing a cheaper approach toenhancing germplasm health and meticulous characterization and evaluation of accessions [51]. For Alliums, which do not mainly reproduce by seed, cryopreservation is the most convenient strategy for their conservation [84]. In onions, this process allows for the preservation of various tissues, such as embryonic axes, shoot tips, or dormant bulbs, using cryoprotectants (Vitrification)to shield them from cellular damage and ice crystal formation [81,85,86]. Cryopreservation enables the storage of germplasm for extended periods, potentially decades or even centuries, whilst maintaining the genetic integrity and viability of the stored materials [81,87]. Cryopreservation of onion pollen, for instance, is an effective strategy for the long-term storage of pollen grains and can be preserved in genebanks as cryobanks. This development has the potential foronion breeders and commercial growers [51].

### 4.4. Field Genebanks

Field genebanks represent an important method of conservation for horticultural species that produce recalcitrant seeds or reproduce by vegetative means. These categories of species are generally not amenable to the conservation of their seeds; rather, they are conserved as live plants in field genebanks [80]. This method entails the cultivation and maintenance of live plants within selected areas [87]. Field genebanks may house a broad range of accessions, including wild relatives and landraces, preserving the germplasm in natural growing conditions. This allows for the preservation, characterization, and evaluation of interesting traits under specific environmental conditions. Such collections necessitate continuous management and monitoring to ensure the survival and integrity of the plants. Onions, particularly those from the aggregatum group such as shallot, which do not produce seeds, are among the crop species that are conserved in field genebanks [51]. For instance, the Genebank of the Leibniz Institute of Plant Genetics and Crop Plant Research (IPK) in Gatersleben, Germany, which maintains one of the world’s largest *Allium* germplasm worldwide, has over 2000 accessions which are permanently maintained in their field genebank [79]. Also, there are vegetatively propagated species of Allium under conservation in the field genebank (FGB) of ICAR-NBPGR Regional stations, including those that are classified as rare, endangered, and threatened [51,88,89,90]. Though the conservation of genetic resources under field genebank conditions requires adequate security and sustainability measures, the method is beneficial in terms of characterization and evaluation, which can easily be conducted on established plants [90].

## 5. Collecting and Conservation Status of Global Onion Genetic Resources

The conservation, cataloging, and utilization of onion germplasm are collaborative endeavors engaged in by a range of institutions, organizations, and private entities across the globe [91]. Regardless of whether these efforts are carried out by national or international repositories, the overarching goal remains the same: to safeguard and exploit the inherent genetic diversity of onions for ongoing and future research, breeding, and cultivation [87,92]. National repositories generally focus on preserving the genetic resources found within their respective countries. They often house a wide variety of onion accessions unique to their regions, thereby maintaining the genetic diversity of onion cultivars within the country [10,93]. These repositories serve as a crucial resource for domestic breeding and research programs. The international repositories have been instrumental in facilitating the exchange and sharing of diverse genetic resources among countries [94]. Under the management of international, regional, and national agricultural institutions, such as CGIAR (Consultative Group on International Agricultural Research) centers, diverse onion accessions sourced from different geographical [94].

## 6. International and National Seed Banks and Germplasm Repositories of Onion

Seed banks and germplasm repositories play an integral role in the collection, preservation, and distribution of onion germplasm, thereby ensuring its availability for conservation, research, or breeding endeavors [39]. Numerous well-established seed banks and germplasm repositories, spanning both national and international levels, maintain extensive collections of onions and other Allium genetic resources (Table 1). International entities such as Biodiversity International and the International Treaty on Plant Genetic Resources for Food and Agriculture (ITPGRFA) significantly contribute to the orchestration of global efforts centered on the collection and conservation of onion genetic resources [95]. These organizations promote international collaboration, facilitate the exchange of germplasm, and support capacity-building initiatives aimed at enhancing the conservation of diverse crops, including onion germplasm [95]. Simultaneously, several nations have initiated their own programs for the collection and conservation of onion genetic resources. These initiatives strive to document and preserve the diversity of onion cultivars, landraces, and wild relatives within their boundaries. To ensure the inclusion of a wide range of onion germplasm, collaborations among research institutes, agricultural organizations, and local communities need to be embraced.

Collaborations at both regional and international levels have been vital in the collection and conservation of onion genetic resources [91]. Key players such as the European Cooperative Program for Plant Genetic Resources (ECPGR) and the Asian Vegetable Research and Development Center (AVRDC) are at the forefront of such conservation initiatives [96]. Their efforts through collaborative projects, workshops, and exchanges of genetic materials foster the pooling of resources and knowledge, thereby enhancing the conservation of onion genetic resources. The collection and conservation of onion genetic resources is a matter of global importance, and various initiatives are being conducted across the world [95]. This includes alliances at the regional level that aim to safeguard the genetic diversity of crops, including onions [82]. These concerted efforts play a critical role in promoting the sustainable production of these vital crops.

### 6.1. Europe

In Europe, the collection and conservation of onions’ and other Alliums’ genetic resources is undertaken by a host of countries and institutions Figure 1 [42,58,97]. One notable organization, the European Cooperative Program for Plant Genetic Resources (ECPGR), facilitates coordination across Europe to safeguard and make use of onion genetic diversity [96]. The Spanish genebank, the Vegetable Germplasm Bank of Zaragoza (BGHZ), harbors large collections of onions, representing a broad source of variability useful for breeding [98]. Also, seeds of Spanish onion landraces collected from the major onion production regions, particularly Galicia, have been maintained at the Centro de InvestigacionesAgrarias de Mabegondo (A Coruña, Spain) [99]. These organizations prioritize the collection and conservation of onion landraces, traditional varieties, and wild relatives that are adapted to regional conditions. Ex situ conservation methods form the cornerstone of onion conservation strategies in Europe [100].

Other significant contributors to onion germplasm conservation in the region include institutions like the Nordic Genetic Resource Center (NordGen) in Scandinavia [101,102]. The Nordic Genetic Resources Center (NordGen) represents a genebank for the Nordic countries (including Denmark, Finland, Iceland, Norway, and Sweden) with the aim to conserve and promote the sustainable use of genetic diversity present in plant, forest, and animal species [65,102,103]. Currently, over 80 accessions of aggregatum (or multiplying) onionshave been collected from home gardens of almost all these Fennoscandian countries and are maintained at the Nordic Genetic Resources Center, mainly under the ex situ system [65]. The VIR (All-Russian Institute of Plant Genetic Resources) is one of the largest genebanksinthe world, harboring a wide variety of Plant Genetic Resources, including onions [104]. VIR genetic resources collection has been carried out throughglobal, regional, international, and inter-institutional cooperation [105]. Currently, the VIR genebank contains over 1000 accessions of Allium species, including onions [29,104].

### 6.2. North America

In North America, a number of institutions and initiatives are involved in the collection and conservation of onion genetic resources, with the most recognized one being the United States National Plant Germplasm System (USDA-NPGS). The USDA-NPGS, managed by the United States Department of Agriculture (USDA), is one of the world’s largest national genebank networks, with a mandate to preserve a diverse range of plant genetics resources for global distribution to researchers [106]. This system encompasses several regional repositories dedicated to the preservation of Plant Genetic Resources, including onions. The NPGS manages over 500,000 accessions of plant germplasm [106], of which over 1800 accessions of onion seeds are under preservation in the Germplasm Repository (Figure 1) situated at Geneva, New Yolk [107]. Also, the Western Regional Plant Introduction Station (WRPIS) of the National Plant Germplasm System (NPGS) collects, conserves, and utilizes agriculturally important plant germplasm resources in the United States [103]. The WRPIS has the mandate to maintain, characterize, evaluate, and distribute Plant Genetic Resources to different categoriesof users, especially for the purposes of research and breeding [107]. Currently, the station maintains over 2600 plant species from 376 genera, including over 1000 accessions of *Allium* genetic resources, which include onions vital for developing commercial varieties [103].

### 6.3. Asia

Asia houses a plethora of significant onion genetic resources under conservation. The Asian Vegetable Research and Development Center (AVRDC), located in Taiwan, plays an integral role in conserving onion genetic resources throughout the continent [91]. Extensive onion germplasm collections have been conducted in various locations in India [108]. India’s National Bureau of Plant Genetic Resources (NBPGR) curates a diverse collection of onion germplasm and actively engages in collaborations with international organizations [51]. The Indian Council of Agricultural Research-Directorate of Onion and Garlic Research (ICAR-DOGR) maintains an array of onion collections, encompassing wild relatives of onions and exotic collections Figure 1 [30,108]. In India, the vast majority of the diverse *Allium* genetic resource collections are mainly composed of common onions and garlic [51]. Other countries in the region, such as South Korea, China, and Japan, have national programs and institutions specifically devoted to the collection and conservation of onion genetic resources [95].

### 6.4. Africa

Onion collections originating from countries in Africa, including Nigeria, Mali, Niger, Egypt, Morocco, Algeria, Sudan, Tunisia, Ethiopia, Kenya, Tanzania, Angola, and South Africa, are reported [10,37,109]. Rouamba and Currah have reported on onion germplasm collections, including landrace sin some African countries such as Burkina Faso (38 accessions at INERA Station de Farako-ba), Niger (16 accessions at Centre’ Regional de RechercheAgricole: CERRA), (C’oted’IvoireFondio L., Institut des Savanes: IDESSA) and Cameroom (Institut de RechercheAgronomique, Centre de Foumbot) [109]. The Natural Resources Institute of the UK has played an instrumental role in promoting onion germplasm collections and conservation in these Africa countries, a project embarked on under the Darwin Initiative program [109]. These initiatives center around preserving local landraces, wild relatives, and commercially significant onion cultivars adapted to the diverse agro-ecological zones in the region. By maintaining this genetic diversity, these institutions play an essential role in safeguarding the sustainability and resilience of onion cultivation in Africa.

### 6.5. South America

In South America, both international and national organizations are actively involved in the collection and conservation of onion genetic resources [110]. They conserve a wide array of onion cultivars and landraces traditionally cultivated in South America. Brazil has set up germplasm banks, and the conservation of onion genetic resources forms a key component of these resources. They engage in regional collaborations aimed at improving the conservation efforts of onion genetic resources, thus ensuring the sustainable use of this essential crop’s genetic diversity. In Brazil, Embrapa (Brazilian Agricultural Research Corporation), a government institution in the country, has played a significant role in introducing and conserving genetic resources in the country [111].

### 6.6. Oceania

For Oceania, the New Zealand Institute of Crop and Food Research is involved in the onion collection introduction, evaluation, and developmentofonion germplasm derived from various sources in order to broaden the base of adapted breeding material [112]. In sum, the task of collecting and conserving onion genetic resources is a global endeavor that spans diverse regions and climates. This global network of conservation efforts operates through collaboration, data sharing, and carefully targeted collection missions. As a result, these initiatives play a critical role in preserving and utilizing the genetic diversity found within onion species. This not only ensures the sustainability of onion production but also paves the way for advancements in onion cultivation and breeding techniques around the world.

**Table 1 plants-12-03294-t001:** List of some major *Allium* genebanks worldwide.

Institute	Acronym	Institute Code	Country	No. of Accessions	Major *Allium* spp.	Reference
ICAR—National Bureau of Plant Genetic Resources (Directorate of Onion and Garlic Research)	ICAR-NBPGR (ICAR-DOR)	IND1457	India	2606	*A. cepa*, *A. sativum*, *A. fistulosum*, *A. ampeloprasum*, *A. chinense*, *A. tuberosum*	[29,51,104]
All-Russian Institute of Plant Genetic Resources	VIR	RUS001	Russia	1888	*Allium*	[29,104]
National Institute of Agrobiological Sciences	NIAS	JPN003	Japan	1352	*Allium*	[90,95,104]
USDA National Plant Germplasm System (NPGS)	USDA-NPGS	USA003	USA	1304	*A. cepa* and wild spp.	[29,51,104]
Leibniz Institute of Plant Genetics and Crop Plant Research	IPK	DEU146	Germany	1264	*A. cepa*, *A. sativum*, *A. fistulosum*, *A. ampeloprasum*, *A. proliferum*, *A. chinense*	[51,104]
AVRDC—The world Vegetable Center	AVRDC	TWN001	Taiwan	1129	*A. ampeloprasum*, *A. sativum*, *A. cepa*, and *A. ascalonicum*	[29,95,104]
Royal Botanic Gardens	RBG	GBR004	UK	1100	*Allium*	[29,104]
The Western Regional (W6) Plant Introduction Station (WRPIS) of the National Plant Germplasm System (NPGS)	W6/WRPIS	USA022	USA	1066	*Allium*	[104]
Science andAdvice for Scottish Agriculture	SASA	GBR165	UK	1005	*Allium*	[104]
Warwick Crop Center/Warwick Genetic Resources Unit	Warwick GRU	GBR006	UK	1755	*Allium*	[29,95]
Research Institute for Plant Production		CZE003	Czech Republic	Unknown	*Allium*	[95]
The Center for Plant Diversity (formally Research Centre for Agrobiodiversity)	CPD (Formally, RCA)	HUN003	Hungary	380	*A. cepa A. sativum*, *ampeloprasum*, *A. fistulosum*	[95]
Center for Genetic Resources	(CGN)	NLD037	The Netherlands	Unkown	*Allium cepa*, *Allium ampeloprasum*, Wild *Allium* species	[95]
Centre for Plant Breeding and Reproduction Research (CPRO-DLO), Department of Vegetable and Fruit Crops, Wageningen	CPRO-DLO	-	The Netherlands	11	*A. ampeloprasum*	[51]
TheNordicgenebank	NorGen	Unknown	Nordic countries	>80	*A. sativum*, welsh onion, potato onion	[65]
Institute for Plant Genetic Resources, Sadovo	IPGR	BGR001	Bulgaria	618	*A cepa*, *A. sativum*	[51]
Plant Genetic Resources Laboratory of Research Institute of Vegetable Crops, Skierniewice	InHort	POL030	Poland	144	*A cepa*	[51]
Crop Research Institute, Prague		Unknown	Czech Republic	157	*A. sativum*	[51]
Vegetable Crop Gene Resources and Germplasm Enhancement, Ministry of Agriculture, Beijing	VCGRGE	Unknown	China	591	*A. sativum*, *A. tuberosum*	[51]
National Agrobiodiversity Center (NAAS), RDA, Suwon	RDA-NAAS	Unknown	Republic of Korea	1158	*A. sativum*	[51]
European Cooperative Programme for Plant Genetic Resources	ECPGR	-	Germany	14,400	*Allium*	[29]

## 7. High-Throughput Phenotyping and Molecular Approaches in Onion Germplasm Characterization and Evaluation

Effective germplasm characterization and evaluation are essential in harnessing the genetic potential of *Allium* crops for breeding, conservation, and sustainable agriculture [113,114]. As advancements in technology and methodologies continue, there are improved methods, including High-Throughput phenotyping systems and molecular techniques that enhance the accuracy, efficiency, and comprehensiveness of *Allium* germplasm characterization and evaluation [113]. This further leads to effective breeding of new genotypes and genetic resources exchange between institutions as well as their distribution to various users.

### 7.1. High-Throughput Phenotyping

The plant phenotype encompasses the plant’s characteristics, including morphology, physiology, and biochemical profiles that reveal the plant’sstructure, composition, and growth [115,116]. It comprises various agronomic traits such as structure, size, and color, as well as seed germination [117]. Conventional phenotyping systems are generally laborious and time-consuming. High-throughput phenotyping methods have already gained much success in the study of many crop species targeting plant traits such as cells, seeds, shoots, leaves, roots, individual plants, and canopy [118,119]. The method involves the application of automated tools such as imaging systems, sensors, and drones for a rapid collection of phenotypic information from a large number of accessions [115,120]. This approach, which is non-destructive in nature, facilitates accelerating the assessment of quantitative traits like plant height, leaf area, and disease resistance, enabling more efficient germplasm evaluation. Remote sensing technologies, such as hyperspectral imaging and thermal imaging systems, have particularly served as non-destructive methods to assess the health, stress, and physiological traits of crops, including onions [120]. The application of these techniques can provide real-time data on plant responses to environmental factors, thus accelerating and simplifying the process of crop diversity assessment.

### 7.2. Molecular Marker Application

Molecular markers are genetic variations in the form of specific DNA sequences that can easily be detected or discovered in the genomes of organisms [121]. By acting as signposts on the genome, DNA sequences provide an understanding ofthe genetic or molecular basis of trait variability, the magnitude of genetic diversity within and between species, as well as the associations among individuals within populations [122]. Molecular markers exist in different forms, with the most popularly used markers being Restriction Fragment Length Polymorphisms (RFLPs) [123], Amplified fragment length polymorphism (AFLP) [124], Random Amplified Polymorphic DNA (RAPD) [125], Simple Sequence Repeats (SSRs) [126], Single Nucleotide Polymorphisms (SNPs) [15], and genotyping-by-sequencing (GBS) [18,29]. These markers have enabledtherapid identification of many distinct genetic profiles, similarities among accessions, and the selection of specific genes or traits of interest [18].

In onion breeding, the application of molecular markers provides a fast means to introgress desirable traits from wild relatives into elite cultivars [110]. Over the past few years, the application of molecular markers as important tools in onion germplasm characterization and genetic diversity conservation has paved the way for a more effective and efficient germplasm conservation management as well as the utilization of the crop’s genetic resources for resilient breeding [125,127,128]. Using molecular markers, many important novel genes have been introduced into edible species of Allium, and this has contributed significantly to increasingthe diversity of the crop [14]. Additionally, the application of molecular markers hasbeen used for mapping important genomic loci on the onion genome, and this has facilitated the use of marker-assisted selection methods to enhance the precision of breeding efforts, such as selecting breeding lines [129,130]. Currently, work focusing on understanding the percentage of duplications or misclassification within and between *Allium* collections requires much research impetus, as the extent of duplication in the crop is still inadequate.

## 8. Gaps in Global Onion Genetic Resources Collection

Despite considerable strides in the collection and conservation of onion genetic resources, some gaps and challenges remain to be tackled [51]. These deficiencies underline the areas requiring additional focus and resources to guarantee a thorough and effective strategy for collecting and conserving the full spectrum of onion genetic diversity.

### 8.1. Underrepresentation of Onion Wild Relatives

Notwithstanding the intuitive value of wild onion relatives as rich sources of genetic diversity for trait exploitation and improvement of cultivated varieties, their collection and conservation efforts have been comparatively limited in some regions of the world [51]. This underscores the need for more targeted strategies to identify, collect, and conserve these wild onion species to harness their genetic diversity and capitalize on their potential contributions to onion breeding programs [131].

### 8.2. Regional and Local Diversity

Germplasm collection efforts atsome regional and local levels often predominantly target commercially significant cultivars or those already used in specific locations [51]. This focus can result in a lack of representation of regional landraces and local onion varieties in germplasm collections [108]. To capture the unique genetic traits and adaptability of these populations, there is a clear need to collect, meticulously document, and conserve the regional and local diversity of onions [109].

### 8.3. Limited Access to Traditional Cultivars

Heirloom or traditional onion varieties handed down through generations and typically conserved by farmers and local communities represent an essential aspect of genetic diversity. Yet, the conservation and accessibility of these traditional cultivars present a challenge due to their localized distribution and underrepresentation in seed banks or germplasm repositories [51]. Thus, enhancing collaboration with local communities to collect and protect these traditional onion cultivars is a necessity.

### 8.4. Conservation of Genetic Erosion Hotspots

Genetic erosion is the loss of genetic diversity over time, and this poses remarkable concern in onion genetic resource preservation efforts. In some regions or communities, there are higher risks of genetic erosion resulting from a variety of reasons, such as changes in farming practices, urbanization, socio-economic factors, and the continual harvesting of wild species [51]. Identifying and targeting these genetic erosion hotspots for collection and conservation is crucial to prevent the loss of unique onion germplasm [130].

### 8.5. Characterization and Evaluation Efforts

The meticulous characterization and evaluation of collected germplasm hold equal importance as the collection process itself [109]. Comprehensive characterization involves evaluating a broad array of traits, including flavor, nutritional content, and adaptability to specific environmental conditions. This detailed knowledge enables researchers to make informed decisions and effectively leverage these genetic resources for the enhancement of onion varieties [110]. In the realm of onions, it has been suggested that numerous germplasm have not yet undergone thorough characterization, and many landraces remain unexplored for potential incorporation into modern breeding programs [18].

### 8.6. Limited Resources for Conservation

The collection and conservation of onion genetic resources, while undeniably beneficial, necessitate substantial resources encompassing funding, infrastructure, and skilled personnel [14,131,132]. Many seed banks, germplasm repositories, and research institutions grapple with financial and capacity constraints [132]. There is a pressing need to bolster resource allocation to achieve efficient conservation of onion germplasm. These resources extend beyond just infrastructure development and include capacity building and sustained funding, all of which are instrumental in bridging the current gaps.

### 8.7. Collaboration and Data Sharing

One crucial gap in onion germplasm management, which applies to other crop germplasm resources as well, is the optimization of collaboration and data sharing among various germplasm repositories, research institutions, and organizations [113]. Inter-institutional and inter-regional collaborations are indispensable to streamline collection and preservation efforts, facilitate germplasm exchange, avoid redundancy, and ensure comprehensive and efficient conservation of onion genetic diversity. A holistic approach that fosters partnerships among research institutions, seed banks, farmers, and international organizations is crucial for addressing the current gaps in global onion genetic resources collection and conservation [28,111]. Such a coordinated system promotes awareness, facilitates targeted collection missions, encourages investment in characterization and evaluation, and aids the effective utilization of onion genetic diversity for sustainable onion production [4,110].

## 9. Countrywide Onion Genetic Resource Collection and Conservation Efforts

Assessment of the status of onion genetic resources collection and conservation activities across various locations within individual countries yields valuable insights into efforts made towards preserving onion diversity. These assessments facilitate the identification of existing gaps, assist in prioritizing conservation activities, and promote the sustainable use of onion genetic resources. By paying close attention to nationwide assessments of onion genetic resources collection and conservation, we can gain a comprehensive understanding of onion diversity, identify conservation priorities, and develop targeted strategies to ensure the preservation and sustainable use of onion genetic resources. Such assessments form the foundation of effective conservation programs and contribute to the resilience, productivity, and cultural heritage associated with onion cultivation.

### 9.1. Inventory of Onion Germplasm

A comprehensive inventory is vital for assessing the breadth of onion genetic resources collected and conserved within a country. This process requires gathering information on existing germplasm collections from seed banks, germplasm repositories, research institutions, and private collections. The inventory should document the number of accessions, categorizing their origin (be it wild relatives, landraces, or cultivars), and should include associated data such as geographical and ecological information, morphological traits, and molecular characterizations, where available.

### 9.2. Documentation of Onion Cultivar Diversity

A nationwide assessment should encompass documentation of the diversity of onion cultivars present within the country. This involves conducting surveys with farmers, seed companies, and local communities to identify and describe traditional onion varieties, landraces, and heirloom cultivars cultivated across various regions. Comprehensive information on their cultivation practices, adaptability to local conditions, flavor profiles, and uses should be collected to capture the full scope of onion genetic resources.

### 9.3. Evaluation of Onion Genetic Erosion

Evaluating the extent of genetic erosion is essential to comprehend the conservation requirements of onion genetic resources. This process entails pinpointing areas or communities where traditional onion varieties face the risk of extinction due to shifts in agricultural practices, urbanization, or socio-economic factors. Surveys and interviews with farmers and local communities can yield insights into the pace and triggers of genetic erosion, thereby enabling the prioritization of conservation initiatives in regions most at risk.

### 9.4. Gap Analysis and Prioritization

The analysis of collected data aids in identifying gaps in the collection and conservation of onion genetic resources. This initiative facilitates the pinpointing of regions, genetic traits, or specific onion varieties that are underrepresented or at imminent risk of loss [130]. Prioritizing these identified gaps enables the development of targeted conservation strategies, which can focus on specific areas for further collection missions, systematic documentation, and dedicated preservation efforts.

### 9.5. Conservation Strategies and Initiatives

Upon a thorough assessment, strategies and initiatives for conservation can be conceptualized and executed. These might entail the establishment or improvement of seed banks, germplasm repositories, or in situ conservation sites specifically for onion genetic resources. It is pivotal to foster collaborations among research institutions, agricultural organizations, and local communities for the successful implementation of these strategies. Additionally, initiatives like public awareness campaigns, farmer education programs, and policy reinforcement to promote seed preservation and sustainable agricultural practices can significantly contribute to onion conservation efforts.

### 9.6. International Collaboration

Although country-wide assessments primarily concentrate on national endeavors, nurturing international collaboration and experience sharing is equally critical [109]. Cooperation with international organizations, genebanks, and regional networks facilitates the exchange of germplasm, knowledge, and best practices [109]. Participation in global initiatives, like the International Treaty on Plant Genetic Resources for Food and Agriculture (ITPGRFA), empowers countries to contribute to and benefit from the broader conservation and utilization of onion genetic resources [95].

## 10. Conclusions, Recommendations, and Future Perspectives

The conservation and global distribution of crop germplasm are essential for ensuring agricultural sustainability and future food security. Onion genetic resources, including landraces, wild relatives, and cultivated varieties, hold valuable traits for crop improvement and market diversification. Preserving and utilizing onion genetic diversity can enhance the resilience of onion production systems, improve crop performance, and address emerging challenges such as climate change, pests, and diseases. To achieve this, targeted collection missions should be undertaken to expand and diversify onion germplasm collections. Special attention should be given to areas with high onion diversity, including regions with unique ecotypes, landraces, and wild relatives. Collaborating with local or indigenous communities and research institutions can strengthen collection efforts and ensure the inclusion of valuable onion genetic resources. Additionally, establishing and strengthening conservation networks and collaborations at regional, national, and international levels is crucial. Sharing germplasm management expertise and best practices among genebanks, research institutions, and local communities can optimize germplasm conservation efforts. Collaborative projects, data-sharing platforms, and capacity-building initiatives will enhance the effectiveness and impact of onion germplasm conservation. In the context of genebank management of onion germplasm, ensuring accurate and comprehensive documentation is essential, as it serves as a prerequisite for effective conservation and utilization.

## Figures and Tables

**Figure 1 plants-12-03294-f001:**
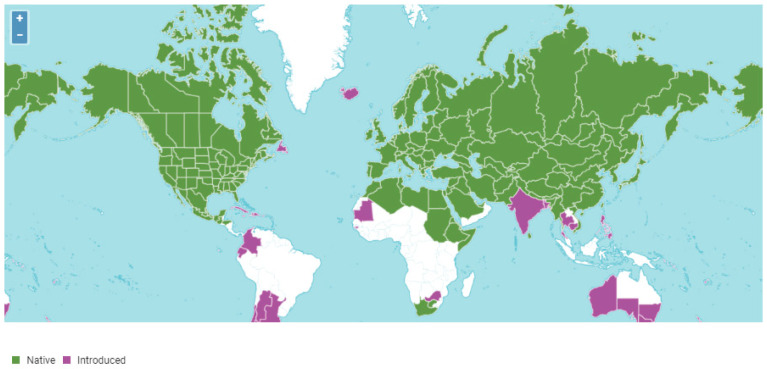
Global distribution of *Allium* species, including onions. Source: Plants of the World Online. https://powo.science.kew.org/ (accessed on 26 July 2023).

## Data Availability

Not applicable.

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
