# Peer review of "Conservation and Global Distribution of Onion (Allium cepa L.) Germplasm for Agricultural Sustainability"

_plants, 2023, doi:10.3390/plants12183294_

Round 1
Reviewer 1 Report
This is an interesting review of the global distribution and protection of onion germplasm resources. As the second largest vegetable after potatoes, the protection of wild relatives, local varieties, and commercial varieties of onions is of great significance for the sustainable development of agriculture. The author provided a detailed introduction to the genetic resources of onions, including their conservation methods and status. The author also emphasized the cooperation between different countries and international organizations and pointed out the gaps in the collection of onion genetic resources. Overall, this ms should be a good comprehensive review of onion germplasm resources. However, this manuscript still needs to be reorganized, especially some figures, tables, and paragraphs that need to be integrated, and some duplicates that need to be deleted.
Some need revisions listed here:
Line 9 delete a diploid species with 16 chromosomes (2n = 2x = 16)
Line 22 the order of the keywords should be changed, as Onion; Diversity; ----.
Line 28 Shallot (Allium cepa L.) What is the difference with onion (Allium cepa L.) in line 27.
Line 43 Fig.1.0--- Fig. 1
Lines 63-58 labeled should be deleted.
Line103 Figure 1.0, The global distribution of onion, here means the total onion genetic resources, if that, some areas like Australia also with cultivated onions. And, in this figure, three groups of onions which may with some specific distribution, it’s better to denote here.
Line 294 Figure 2 Although the figure source is indicated, it still appears inappropriate as the main image here.
Line 326 Table 1. Here list of some major Allium gene banks, what is a proportion of onions in it, and that is regrettable.
Lines 228-243 should be integrated into the onion germplasm conservation methods part.
This is an interesting review of the global distribution and protection of onion germplasm resources. As the second largest vegetable after potatoes, the protection of wild relatives, local varieties, and commercial varieties of onions is of great significance for the sustainable development of agriculture. The author provided a detailed introduction to the genetic resources of onions, including their conservation methods and status. The author also emphasized the cooperation between different countries and international organizations and pointed out the gaps in the collection of onion genetic resources. Overall, this ms should be a good comprehensive review of onion germplasm resources. However, this manuscript still needs to be reorganized, especially some figures, tables, and paragraphs that need to be integrated, and some duplicates that need to be deleted.
Some need revisions listed here:
Line 9 delete a diploid species with 16 chromosomes (2n = 2x = 16)
Line 22 the order of the keywords should be changed, as Onion; Diversity; ----.
Line 28 Shallot (Allium cepa L.) What is the difference with onion (Allium cepa L.) in line 27.
Line 43 Fig.1.0--- Fig. 1
Lines 63-58 labeled should be deleted.
Line103 Figure 1.0, The global distribution of onion, here means the total onion genetic resources, if that, some areas like Australia also with cultivated onions. And, in this figure, three groups of onions which may with some specific distribution, it’s better to denote here.
Line 294 Figure 2 Although the figure source is indicated, it still appears inappropriate as the main image here.
Line 326 Table 1. Here list of some major Allium gene banks, what is a proportion of onions in it, and that is regrettable.
Lines 228-243 should be integrated into the onion germplasm conservation methods part.
Author Response
Dear Reviewers,
We sincerely thank you for your constructive comments. We are sure the quality of the manuscript has improved based on your recommendations and we are grateful for your time and effort. We have amended the manuscript to the best possible standard, keeping in mind your comments and suggestions. Thank you.
Sincerely,
Dr. Seong-Hoon Kim
Review 1
Review comment: Line 9 delete a diploid species with 16 chromosomes (2n = 2x = 16)
Author response: Thank you for drawing our attention to this. We have deleted the statement “a diploid species with 16 chromosomes (2n = 2x = 16)” in Line 9 of the abstract section.
Review comment: Line 22 the order of the keywords should be changed, as Onion; Diversity; ----.
Author response: In our manuscript, the order of the key words was arranged in an alphabetical order. However, following your recommendation, we have changed the order to “Key word: Onion; Diversity; Germplasm; Genebank; Climate change”
Review comment: Line 28 Shallot (Allium cepa L.) What is the difference with onion (Allium cepa L.) in line 27.
Author response: Thank you for drawing our attention to the mistake made where we repeated the scientific name of onion which is Allium cepa for that of shallot. We have made the correction and we now have the scientific name of shallot as “Allium ascalonicum L.”.
Review comment: Line 43 Fig.1.0--- Fig. 1
Author response: In Line 43, we have changed “Fig. 1.0” to Fig. 1 as suggested.
Author response: Lines 63-58 labeled should be deleted.
Author response: Please, this is unclear. I could not find the label indicated to be deleted.
Review comment: Line103 Figure 1.0, The global distribution of onion, here means the total onion genetic resources, if that, some areas like Australia also with cultivated onions. And, in this figure, three groups of onions which may with some specific distribution, it’s better to denote here.
Author response: Thank you for this observation. We have modified figure 1. The modified version encompasses Allium species in general.
Review comment: Line 294 Figure 2 Although the figure source is indicated, it still appears inappropriate as the main image here.
Author response: Thank you for this suggestion. We have removed figure 2 from our revised manuscript.
Review comment: Line 326 Table 1. Here list of some major Allium gene banks, what is a proportion of onions in it, and that is regrettable.
Author response: Thank you for this suggestion. Currently, we do not have adequate and comprehensive information on the total number of onion genetic resources under conservation for all the genebank, so in our revised version of the manuscript, we have attempted to specify the main Allium species being reported from the various gene banks.
Review comment: Lines 228-243 should be integrated into the onion germplasm conservation methods part.
Author response: As recommended, we have integrated Line 228 – 243 into the subsection “onion germplasm conservation methods” in Line 244-255

Reviewer 2 Report
The review article is well written except for few minor changes to be addressed as mentioned below:
(1) A table presenting the genetic, morphological characteristics of different species of Allium can be presented
(2) The Headings 2.2 and 3.0 are similar, hence to be modified
Author Response
Dear Reviewers,
We sincerely thank you for your constructive comments. We are sure the quality of the manuscript has improved based on your recommendations and we are grateful for your time and effort. We have amended the manuscript to the best possible standard, keeping in mind your comments and suggestions. Thank you.
Sincerely,
Dr. Seong-Hoon Kim
Review 2
Review comment: A table presenting the genetic, morphological characteristics of different species of Allium can be presented
Author response: Thank you for this recommendation. Currently, we have limited time to revise our manuscript. In our next manuscript, we will provide a full report on the genetic and morphological characteristics of different species of Allium.
Review comment: The Headings 2.2 and 3.0 are similar, hence to be modified
Author response: Thank you for drawing our attention to the headings for subsections 2.2 and 3.0 which are the same. We have changed the heading in subsection 3.0 to “Groups of Onions”

Reviewer 3 Report
The ms plants-2568633 with the title of Conservation and Global Distribution of Onion (Allium cepa L.) Germplasm for Agricultural Sustainability investigates and reviews an important topic, but the authors must improve the ms before it can be accepted in such high quality journal, otherwise the ms should be rejected.
Authors should follow the format of the journal in citing the references within the text and in the list of references.
The authors must cite any text that was not their own work.
The authors should add additional investigations that were recently published.
L9 remove following text: , a diploid species with 16 chromosomes (2n = 2x = 16),
L20 use another word instead of furnish
L29 Allium wakegi should be italic
L25-38 please revise this paragraph and remove the common text.
L106-115 please add suitable citations for this text
L118-123 please add suitable citations for this text, authors must cite any text since such text is not their own work.
L210-289 I can not believe that these paragraphs are written without citations. This issue must be revised, otherwise, the ms should be rejected.
Fig. 2.0. Allium field collection maintained at IPK, in Gatersleben, Germany (Source: Panis et al [39]. The authors should have permission to use such Figures in their ms.
L300-325 please add suitable citations for this text, authors must cite any text since such text is not their own work.
Table 1 should be revised using solid references
L331-359 please add suitable citations for this text
Conclusion, Recommendations and Future Perspectives, although the authors covered most of topic but I suggest the authors to make this section shorter and write what they want to tell the readers straight.
Using 48 citations and most of them are general references is not enough to get your review ms published in such journal.
Good luck
Major editing is needed
Author Response
Dear Reviewers,
We sincerely thank you for your constructive comments. We are sure the quality of the manuscript has improved based on your recommendations and we are grateful for your time and effort. We have amended the manuscript to the best possible standard, keeping in mind your comments and suggestions. Thank you.
Sincerely,
Dr. Seong-Hoon Kim
Review 3
Review comment: Authors should follow the format of the journal in citing the references within the text and in the list of references.
Author response: We have modified our manuscript according to the journal and also provided adequate reference citations and the reference list presented following the instruction of the journal.
Review comment: The authors must cite any text that was not their own work.
Author response: Thank you for drawing our attention to this. We have provided citations appropriate to the text in the manuscript.
Review comment: The authors should add additional investigations that were recently published.
Author response: Thank you. Our revised version of the manuscript contains many current references.
Review comment: L9 remove following text: , a diploid species with 16 chromosomes (2n = 2x = 16),
Author response: Thank you for drawing our attention to this; we have deleted “a diploid species with 16 chromosomes (2n = 2x = 16)” in Line 9 of the abstract section as suggested.
Review comment: L20 use another word instead of furnish
Author response: In L20, we have removed the word ‘furnished’ as it appeared in the sentence “ In this review, we furnish an exhaustive discussion on the preservation and worldwide distribution of onion germplasm…”
Review comment: L29 Allium wakegi should be italic
Author response: Thank you for this observation. The species name of ‘scallion’ has been italicized and we now have “scallion (Allium wakegi).
Review comment: L25-38 please revise this paragraph and remove the common text.
Author response: The paragraph in L25 – 38 has been modified with the common text detected being removed. The word ‘worldwide’ is deleted. Also ……….”during the era of ancient human civilization” is deleted.
Review comment: L106-115 please add suitable citations for this text.
Author response: Thank you. We have provided citations for the text in L106 – 115
Review comment: L118-123 please add suitable citations for this text, authors must cite any text since such text is not their own work.
Author response: Thank you for drawing our attention to the limited number of citations provided in L118-123. We have rived our manuscript and added suitable citations in L118 – 123.
Review comment: L210-289 I can not believe that these paragraphs are written without citations. This issue must be revised, otherwise, the ms should be rejected.
Author response: Thank you for drawing our attention once again to the limited number of citations provided in L210 – 289. We have rived our manuscript and added suitable citations to L210 – 289.
Review comment: Fig. 2.0. Allium field collection maintained at IPK, in Gatersleben, Germany (Source: Panis et al [39]. The authors should have permission to use such Figures in their ms.
Author response: Thank you for your suggestion. We have removed figure 2 from our revised manuscript.
Review comment: L300-325 please add suitable citations for this text, authors must cite any text since such text is not their own work.
Author response: Thank you for this observation and suggestion. We have provided suitable citations to the text in L300 – 325
Review comment: Table 1 should be revised using solid references
Author response: Thank you for your suggestion. We have revised Table 1, with references provided.
Review comment: L331-359 please add suitable citations for this text
Author response: Thank you for this observation and suggestion. We have provided suitable citations to the text in L331 – 359
Review comment: Conclusion, Recommendations and Future Perspectives, although the authors covered most of topic but I suggest the authors to make this section shorter and write what they want to tell the readers straight.
Author response: The message provided in the “Conclusion, Recommendations and Future Perspectives” section has been modified (shortened).
Review comment: Using 48 citations and most of them are general references is not enough to get your review ms published in such journal.
Author response: We have revised the entire manuscript and provided additional references to support and strengthen statements made in the text.

Reviewer 4 Report
The authors present a review on the present conservation of onion genetic resources, including identified gaps and targets for future efforts. By and large, the paper is well written and structured. There only a few things to amend.
L52 -57 In the current climate,.... genes and gene pools [19]
this section is repeated word by word from L63 -68- I assume an error of copy & paste that should be corrected
L101 Figure 1.0
The meaning of this figure is not clear to me. The map indicates a center of origin (green), and many countries / regions where onions have been introduced (red). What about many of the white areas? The authors themselves state earlier that Turkey (white) is one of the biggest producers, other countries are white as well (Germany, Austria, Hungary to name but a few in Europe) while having a sizable Allium cepa acreage. At least for the Euopean countries, "white" can't mean lack of data, so this figure needs to be explained better, or left out.
L326 Table 1
The hyperlink to the IPK institute should read ...ipk-gaterslebe.de (not gertersleben)
The review is generally quite comprehensive and well written. I would have liked to see though a short discussion on what role (if any) molecular markers can have in interlinking various efforts between countries and institutions, eg in identifying doubling of accessions or gaps in preserved genetic diversity. Recording of molecular data is mentioned by the authors in passing, but not discussed any further in its merits or limitations.
Author Response
Dear Reviewers,
We sincerely thank you for your constructive comments. We are sure the quality of the manuscript has improved based on your recommendations and we are grateful for your time and effort. We have amended the manuscript to the best possible standard, keeping in mind your comments and suggestions. Thank you.
Sincerely,
Dr. Seong-Hoon Kim
REVIEWER 4
Review comment: L52 -57 In the current climate,.... genes and gene pools [19]
this section is repeated word by word from L63 -68- I assume an error of copy & paste that should be corrected
Author response: Thank you for drawing our attention to the word by word repetition made in Lines 52 – 57 and Line 63 – 68 in the introduction section of our manuscript. The repetition in line 63 – 68 has been deleted from paragraph 3 of the introduction section.
Review comment :L101 Figure 1.0
The meaning of this figure is not clear to me. The map indicates a center of origin (green), and many countries / regions where onions have been introduced (red). What about many of the white areas? The authors themselves state earlier that Turkey (white) is one of the biggest producers, other countries are white as well (Germany, Austria, Hungary to name but a few in Europe) while having a sizable Allium cepa acreage. At least for the Euopean countries, "white" can't mean lack of data, so this figure needs to be explained better, or left out.
Author response: Thank you for this observation. We have modified figure 1. The modified version encompasses Allium species in general which are distributed worldwide.
Review comment: L326 Table 1
The hyperlink to the IPK institute should read ...ipk-gaterslebe.de (not gertersleben)
Author response: This figure has been modified and the hyperlink no longer exists
Review comment: The review is generally quite comprehensive and well written. I would have liked to see though a short discussion on what role (if any) molecular markers can have in interlinking various efforts between countries and institutions, eg in identifying doubling of accessions or gaps in preserved genetic diversity. Recording of molecular data is mentioned by the authors in passing, but not discussed any further in its merits or limitations.
Author response: Thank you for this suggestion. We have provided a brief discussion on molecular marker application in onion germplasm characterization and evaluation. Currently, we have limited time to complete our revision, in future; we will provide a full review work on molecular marker applications in Allium genetic resource characterization and evaluation.

Round 2
Reviewer 1 Report
The onion is the second largest vegetable crop in global production by value, surpassed only by the tomato. The revision of the manuscript has indeed improved a lot, but there are still some minor issues, especially regarding the integration of different paragraph content. In addition, the onion genome article published in 2022, is of great significance for the utilization and innovation of onion genetic resources. It is recommended that the author cite this literature. Line 20: discussedthe--- discussed the Line 105: Figure 1.0.---Figure 1. Line 145: A. vavilovii should be italicized. Line229: over 20000? ---over 20000 accessions? Line 228 3.0.---3. Also, the groups of onion--- Infraspecific classification of onion Line 231 here the ref.38 may be not correct. Please check Fritsch R M & Friesen N (2014) 1 Evolution, Domestication and Taxonomy. You can download it from the web and in this paper, a group conception was first suggested. Line 271: 5.1. line 286 5.2, line 297 5.3, line 318 5.4. All these four paragraphs are the content of ex situ conservation, so they should be listed under 4.1, and 4.2. 4.3, & 4.4. Same as above, 7.0. (7.1-7.5) should be moved to 6.0 (6.1-6.5) Line 480, all the Allium species names in the table should be in italics. Line 485-537: same as above. Also, Liao et al. (2022) published their article on Allium crops, It is a very important contribution to understanding the onion genome evolution and flavor formation.The onion is the second largest vegetable crop in global production by value, surpassed only by the tomato. The revision of the manuscript has indeed improved a lot, but there are still some minor issues, especially regarding the integration of different paragraph content. In addition, the onion genome article published in 2022, is of great significance for the utilization and innovation of onion genetic resources. It is recommended that the author cite this literature. Line 20: discussedthe--- discussed the Line 105: Figure 1.0.---Figure 1. Line 145: A. vavilovii should be italicized. Line229: over 20000? ---over 20000 accessions? Line 228 3.0.---3. Also, the groups of onion--- Infraspecific classification of onion Line 231 here the ref.38 may be not correct. Please check Fritsch R M & Friesen N (2014) 1 Evolution, Domestication and Taxonomy. You can download it from the web and in this paper, a group conception was first suggested. Line 271: 5.1. line 286 5.2, line 297 5.3, line 318 5.4. All these four paragraphs are the content of ex situ conservation, so they should be listed under 4.1, and 4.2. 4.3, & 4.4. Same as above, 7.0. (7.1-7.5) should be moved to 6.0 (6.1-6.5) Line 480, all the Allium species names in the table should be in italics. Line 485-537: same as above. Also, Liao et al. (2022) published their article on Allium crops, It is a very important contribution to understanding the onion genome evolution and flavor formation.
Author Response
Dear Reviewers,
Your insightful feedback has been invaluable in refining our manuscript. Assuredly, its caliber has been elevated in light of your astute recommendations. We've meticulously revised it to the pinnacle of standards, bearing in mind your invaluable suggestions. Your dedication and time are deeply appreciated.
Sincerely,
Dr. Seong-Hoon Kim
REVIEWER 1
Comments and Suggestions for Authors
Comments and Suggestions: The onion is the second largest vegetable crop in global production by value, surpassed only by the tomato. The revision of the manuscript has indeed improved a lot, but there are still some minor issues, especially regarding the integration of different paragraph content. Some need revisions listed here:
Author response: We are very grateful for your comments. Authors have resolved all issues related to the paragraph contents. Where needed, we have integrated some paragraph contents.
Comments and Suggestions: In addition, the onion genome article published in 2022, is of great significance for the utilization and innovation of onion genetic resources. It is recommended that the author cite this literature.
Author response: Thank you for recommending this article for our consideration. We have read through the reference on the onion genome article published in 2022 and is now cited in our manuscript.
Comments and Suggestions: Line 20: discussed the--- discussed the
Author response: The authors are grateful for this observation made. The words ‘discussed’ and ‘the’ have been separated and no longer exist as ‘discusedthe’
Comments and Suggestions: Line 105: Figure 1.0.---Figure 1.
Author response: As suggested, in Line 105, ‘Figure 1.0.’ has been changed to ‘Figure 1.’
Comments and Suggestions: Line 145: A. vavilovii should be italicized.
Author response: Thank you for your comment and pointing to this oversight. The authors have italicized ‘A. vavilovii’ and we now have “A. vavilovii” in the text.
Comments and Suggestions: Line 229: over 20000? ---over 20000 accessions?
Author response: The authors are grateful for your observation on the omission of the word ‘accession’. The omitted word has been corrected and we now have “…over 20000 accessions” instead of ‘20000’
Comments and Suggestions: Line 228 3.0.---3. Also, the groups of onion--- Intraspecific classification of onion
Author response: Thank you for this important suggestion. In Line 228, the subheading ‘3.0’ has been changed from “Groups of onion” to “Intraspecific classification of onion”
Comments and Suggestions: Line 231 here the ref. 38 may be not correct. Please check Fritsch R M & Friesen N (2014) 1 Evolution, Domestication and Taxonomy. You can download it from the web and in this paper, a group conception was first suggested.
Author response: Thank you for this observation. Reference 38 has been corrected. As indicated by the reviewer, the right reference is “Fritsch, R. M., and N. Friesen, 2002: Evolution, domestication and taxonomy. In: H. D. Rabinowitch, and L. Currah (eds), Allium Crop Science: Recent Advances, 5—30. CAB International, Wallingford, UK.”. In the reference section of the manuscript, this reference appears ‘[61]’. Following this, the original reference numbers 61 - 75, have been changed to 62 – 76 while the original reference number [76] is removed from the list.
Comments and Suggestions: Line 271: 5.1. line 286 5.2, line 297 5.3, line 318 5.4. All these four paragraphs are the content of ex situ conservation, so they should be listed under 4.1, and 4.2. 4.3, & 4.4.
Author response: Thank you for this observation. As observed, Line 271: 5.1., line 286 5.2, line 297 5.3, and line 318 5.4 have been corrected and renumbered as 4.1, 4.2. 4.3 and 4.4. Now line 340 (6.0) has been renumbered as 5.0.
Comments and Suggestions: Same as above, 7.0. (7.1-7.5) should be moved to 6.0 (6.1-6.5)
Author response: Authors are grateful for your comments. In line 357, subheading 7.0 has been changed to 6.0, hence subheadings 7.1 – 7.6 has been renumbered as 6.1, 6.2, 6.3, 6.4, 6.5 and 6.6.
Comments and Suggestions: Line 480, all the Allium species names in the table should be in italics.
Author response: All the Allium species named in Table 1 have been italicized as indicated by the reviewer. This, as occurred in the manuscript was an oversight.
Comments and Suggestions: Line 485-537: Same as above.
Author response: From Line 485-537, the subheadings 8.0 and 8.1 –8.2 have been renumbered as 7.0 and 7.1-7.2.
***Thus, subheadings in:
- Line 539 - 605 (9.0 and 9.1-9.7), have been renumbered as 8.0 and 8.1 – 8.7.
- Line 607 – 665 (10.0 and 10.1 – 10.6) have been changed to 9.0 and 9.1 – 9.6
- Line 666, the subheading 11.0 has been changed to 10.0
Comments and Suggestions: Also, Liao et al. (2022) published their article on Allium crops, It is a very important contribution to understanding the onion genome evolution and flavor formation.
Author response: Thank you for recommending this article for our consideration. We have read through the reference on the onion genome article published in 2022 and is now cited in our manuscript as found in the reference number [135].

Reviewer 3 Report
The ms has been improved, but the authors should recheck the citations within the text with the list of references and revise this issue.
Moderate editing of English language required
Author Response
Dear Reviewers,
Your insightful feedback has been invaluable in refining our manuscript. Assuredly, its caliber has been elevated in light of your astute recommendations. We've meticulously revised it to the pinnacle of standards, bearing in mind your invaluable suggestions. Your dedication and time are deeply appreciated.
Sincerely,
Dr. Seong-Hoon Kim
REVIEWER 3
Comments and Suggestions for Authors
Comments and Suggestions: The MS has been improved, but the authors should recheck the citations within the text with the list of references and revise this issue.
Author response: Thank you for this suggestion. We have rechecked carefully all reference citations within the text, paying particular attention to their corresponding list of references. Now, we are certain of no further issues.
Comments and Suggestions: Comments on the Quality of English Language: Moderate editing of English language required
Author response: Thank you for your recommendation. We read carefully through the manuscript and endeavored to improve on the quality of English Language. Currently, we consider that our manuscript is in good shape in terms of quality of English.
